# Connecting the SYK Dots

**Dmitri V. Khveshchenko**

Department of Physics and Astronomy, University of North Carolina, Chapel Hill, NC 27599, USA;
khvesh@physics.unc.edu

**Abstract:** We study a putative (strange) metal-to-insulator transition in a granular array of the Sachdev–Ye–Kitaev (SYK) quantum dots, each occupied by a large number $N \gg 1$ of charge-carrying fermions. Extending the previous studies, we complement the SYK couplings by the physically relevant Coulomb interactions and focus on the effects of charge fluctuations, evaluating the conductivity and density of states. The latter were found to demonstrate marked changes of behavior when the effective inter-site tunneling became comparable to the renormalized Coulomb energy, thereby signifying the transition in question.

**Keywords:** granular system; metal-insulator transition; SYK model

## 1. Introduction

The recent upsurge of interest in the SYK and related models of (super) strongly interacting dispersionless fermions with all-to-all $q$-body couplings has been driven, among other reasons, by the hopes of utilizing them as (asymptotically) solvable examples of the so-called non-Fermi liquid (NFL) behavior.

Originally, the SYK reincarnation [1–3] of the parent SY (Sachdev-Ye) [4–8] model was formulated in terms of neutral Majorana fermions that would be abundantly present in the recent theoretical constructs (albeit less so in their attempted experimental realizations). However, in order to account for the physically relevant charge (alongside energy) dynamics, one needs to use charged (complex or Dirac) fermions [9,10].

Regardless of the nature of its constituent fermions, though, the original SYK model lacks any spatial dispersion, and therefore, can be best thought of as a (zero-dimensional) "quantum dot." As such, this system exhibits a characteristic "local NFL" behavior characterized by the anomalous power-law decay of its temporal (but not spatial) correlations [1–8].

Therefore, while predicting some markedly novel features [11–15] in mesoscopic charge and heat transport through its proposed (but not yet implemented) realizations in the irregularly shaped graphene flakes, multi-connected Quantum Hall setups, semiconductor wires and quantum dots, and topological insulator surfaces [16–20], the complex SYK model has still to be extended into the spatial dimensions before applying it to the analysis of any documented higher-dimensional NFL system.

In the early "SYK-lattice" constructions [21–35], the individual SYK dots would be arranged in a regular array by adding short-range (nearest-neighbor) one- and/or two-body entangling terms into the Hamiltonian. Alternatively, the immobile SYK fermions would be hybridized with their conduction counterparts or subject to long-range and distance-dependent many-body couplings.

Such generalizations allow for a variety of the NFL regimes, some of which are even capable of ostensibly reproducing, e.g., the ubiquitous linear temperature dependence of electrical resistivity [36–39].

Still different is a formulation of the SYK model directly in the momentum space which approach appears to be miraculously successful in providing nothing short of a quantitative agreement between

the computed linear resistivity and its measured values in a sizeable number of the well-studied experimental compounds [40].

Among the central issues studied in the context of the SYK-lattices are putative phase transitions between the parent NFL SYK state (often referred to as "strange metal" (SM)) and a more conventional (disordered) Fermi liquid (FL), or alternatively, a (many-body) Mott insulator (MI).

However, the previous analyses were, by and large, limited to the effects of the (somewhat exotic) SYK-type entangling correlations, thereby leaving out the far more mundane (yet physically relevant and practically unavoidable) charge couplings of the Coulomb origin that are going to affect any feasible solid-state implementations of the SYK system, including those of [16–20].

More specifically, in such electron-based simulations the $SYK_4$ interaction itself would be simulated through the geometrically randomized intra-site Coulomb couplings. That alone makes it anything but consistent to neglect the (non-random) inter-site charging effects, if a viable SYK-lattice were to be engineered out of the single-site SYK building blocks akin to those proposed in [16–20].

In defense of the earlier studies of [21–35], concrete practical realizations of the SYK-lattices did not seem to be particularly high on their agenda, while any speculations regarding their potential applications to such long-standing experimental challenges as the high-$T_c$ cuprates or heavy-fermion materials [36–38,40] would be made largely as a matter of custom.

In the present note, we fill in the gap by investigating the charging effects in a manner similar to that utilized in the context of the ordinary (FL) granular electronic materials [41–48].

## 2. SYK Model of Charged Fermions

The Hamiltonian of the SYK array can be written in terms of the complex fermions $\psi_{i\alpha}$ localized at the dot $i$ and carrying a flavor $\alpha = 1, \ldots, q$ (hereafter $q$ is an even integer)

$$H = H_{SYK} + H_T + H_C \tag{1}$$

whose sum includes the customary SYK intra-site $q$-fermion couplings and a chemical potential $\mu$

$$H_{SYK} = \sum_{i;\alpha,\ldots\beta} J_i^{\alpha\ldots\beta} \psi_{i\alpha}^\dagger \ldots \psi_{i\beta} - \mu \psi_{i\alpha}^\dagger \psi_{i\alpha} \tag{2}$$

as well as inter-site tunneling

$$H_T = \sum_{ij;\alpha\beta} t_{ij}^{\alpha\beta} \psi_{i\alpha}^\dagger \psi_{j\beta} \tag{3}$$

and both intra and inter-site charging energies

$$H_C = \sum_{i,j;\alpha,\beta} \frac{U_{ij}}{2} (\psi_{i\alpha}^\dagger \psi_{i\alpha} - Q_i)(\psi_{j\beta}^\dagger \psi_{j\beta} - Q_j) \tag{4}$$

which include the offset charges $Q_i$, if any (in units of electron charge).

In the SYK model the $q$-fermion amplitudes $J_i^{\alpha\ldots\beta}$ in (2) are treated as Gaussian random variables with the time and state-independent variances

$$< J_i^{\alpha\ldots\beta} J_j^{\alpha'\ldots\beta'} >= \frac{J^2}{N^{q-1}} \delta_{ij} \delta^{\alpha\alpha'} \ldots \delta^{\beta\beta'} \tag{5}$$

Averaging (1) over such distribution results in introducing temporally bi-local $2q$-fermion terms to the effective action [1–8].

Likewise, in most of the previous studies of the SYK-lattices [21–35] the tunneling amplitudes in (3) would be treated as random, with the dispersion

$$< t_{ij}^{\alpha\beta} t_{ij}^{\alpha'\beta'} > = \frac{t^2}{N} \delta_{j,i+\hat{\mu}} \delta^{\alpha\alpha'} \delta^{\beta\beta'} \tag{6}$$

where $\hat{\mu}$ is one of $z$ (coordination number) prime vectors of the SYK lattice. Upon averaging, the tunneling term would then result in the inter-site $SYK_2$-type coupling.

Alternatively, one might view such amplitudes as fixed at some $N$-independent value and diagonal in the "flavor" space, $t_{ij}^{\alpha\beta} = t\delta_{ij}\delta^{\alpha\beta}$.

Lastly, the offset charges can also be thought of as random variables; the degree of their disorder ranges from strong (described by a uniform distribution within the entire interval $-1/2 < Q_i < 1/2$, whose situation might be appropriate for naturally assembled networks) to weak (confining the charges to certain values; e.g., $|Q_i| \ll 1$, more suitable for artificially patterned arrays).

A systematic investigation into those different situations would definitely be warranted. However, in much of what follows we drop the offset charges altogether, focusing on the regimes that are farthest from (accidental) degeneracies. In that regard, our main goal is to demonstrate that a conducting state could emerge even under the least conducive conditions.

To that end, the previous studies of the hybrid model with the intra-site $SYK_4$ and inter-site $SYK_2$ couplings—but without any charging effects—have repeatedly reported observing a crossover from the SM described by the ergodic $SYK_4$ model to a disordered FL state corresponding to the non-ergodic $SYK_2$ one at temperatures of order the effective fermion kinetic energy $t^* \sim t^2/J$ [21–35]. However, a potentially critical impact of the Coulomb blockade (CB) due to the charging energy (4) has not been investigated.

By contrast, in the conventional (FL) granular arrays the latter has long been known to invariably drive the system insulating [41–48]. On the other hand, a coupling to some dissipative sub-Ohmic bath was shown to quell the CB, thereby promoting a conducting state [41–44].

Below, we demonstrate that in the problem at hand the role of such sub-Ohmic bath is played by the intra-site SYK correlations themselves, thereby enabling the metal-to-insulator transition (MIT) in the granular SYK systems even in the presence of the charging effects.

## 3. SYK Strange Metal

First, we consider the $U \to 0$ limit where the influence of the tunneling term on the on-site fermion propagator

$$G_i(\tau) = \frac{1}{N} \sum_\alpha < \psi_{i\alpha}(\tau) \psi_{i\alpha}^\dagger(0) > = (\partial_\tau - \mu - \Sigma_i)^{-1} \tag{7}$$

where is captured in terms of the intra-site self-energy

$$\Sigma_i(\tau) = J^2 G_i^{q-1}(\tau) + \sum_j t_{ij} G_j(\tau) t_{ji} \tag{8}$$

where the first term represents the effect of the SYK correlations. This approximation can be further improved, thereby systematically recovering all the (even order) tunneling processes.

For $t, U \to 0$ and $N \gg 1$ the fermion propagator (7) takes the spatially local SYK form

$$G_{ij}(\tau) = \delta_{ij} G_i(\tau) = \delta_{ij} A sgn(\tau) e^{sgn(\tau)\pi\mathcal{E}} / (J\tau)^{2\Delta} \tag{9}$$

where $\Delta = 1/q$, the prefactor $A$ is a known function of $q$, and the dimensionless parameter $\mathcal{E}(\mu)$ controls the fermion density [1–3].

Apart from the mean-field solution (9), in the no-tunneling/zero-charging energy limit the theory (1) possesses a manifold of nearly degenerate solutions which are continuously connected to (9) by

virtue of arbitrary diffeomorphisms of the thermodynamic time variable $\tau \to f_i(\tau)$, obeying the boundary conditions $f_i(\tau + 1/T) = f_i(\tau) + 1/T$, combined with local $U(1)$ phase rotations [1–3,9,10].

$$G_i(\tau_1, \tau_2) = A e^{i\Phi_i(\tau_1) - i\Phi_i(\tau_2)} \left( \frac{\partial_\tau f_i(\tau_1) \partial_\tau f_i(\tau_2)}{(f_i(\tau_1) - f_i(\tau_2))^2} \right)^\Delta \tag{10}$$

In particular, a finite-temperature counterpart of (9) can be obtained by the conformal mapping onto a thermal circle, $\tau \to \sin(\pi T \tau)/\pi T$.

In addition to being spontaneously broken by the particular choice of the mean-field solution (9) down to the subgroup formed by the Mobius transformations $SL(2, R)$, the reparametrization symmetry gets violated explicitly by the temporal gradients $\partial_\tau f$, and the tunneling and Coulomb terms in Equation (1).

Importantly, the $U(1)$ phase fluctuations have no effect on the intra-site SYK terms, whereas the inter-site tunneling terms can be heavily impacted.

As shown in the earlier studies of the SYK-lattices, the low-energy collective charge and energy fluctuations about the mean-field solution (9) can propagate even in the absence of a bare single-particle dispersion ($< t_{ij} > = 0$), as manifested by the same-site localization of the fermion propagator (9).

Small fluctuations are governed by the Gaussian action

$$S_G(\delta\Phi, \delta f) = \sum_{\mathbf{q}} \int_\omega \left( \frac{1}{2E_C} |\delta\Phi|^2 |\omega| (|\omega| + \mathcal{D}\mathbf{q}^2) \right.$$
$$\left. + \frac{\gamma N}{2J} |\delta f|^2 |\omega| (|\omega| + \mathcal{D}'\mathbf{q}^2)(\omega^2 - (2\pi T)^2) \right) \tag{11}$$

where $\gamma$ is a $q$-dependent coefficient vanishing for $q = 2$ [9,10] and the momentum sum goes over the Brillouin zone of the SYK lattice.

The diffusion coefficients $\mathcal{D}$ and $\mathcal{D}'$ pertain to the spatial spreading of charge and energy, respectively. Their values are expected to comply with the lower bound of order $(t^* a)^2/T$ (here $a$ is the lattice constant) in the high-$T$ regime where the inelastic SYK scattering becomes the fastest equilibration mechanism [9,10].

The phase fluctuations $\delta\Phi$ described by the first term in (11) develop below the (independent of $N$) charging energy $E_C$, which alongside the intra-/inter-site capacitive couplings, includes the energy of induced voltages, $E_C^{-1} = U^{-1} + (\partial Q/\partial\mu)_T$, the second term being due to the fermion compressibility.

In turn, the second term in (11) describes the low-energy dynamics of the SYK reparametrization mode and originates from the intra-site Schwarzian derivative $Sch\{\tan \pi T f, \tau\}$ defined as follows: $Sch\{y, x\} = (y'''/y') - (3y''/2y')^2$ [1].

The changing of variables $\partial_\tau f_i = e^{\phi_i}$ yields, in addition to the quadratic term in Equation (11), the non-Gaussian ("Liouville") interaction $S_{NG}(\delta f) = (2\pi T)^2 \frac{\gamma N}{J} \sum_i \int_\tau e^{2\phi_i(\tau)}$ [1–3].

Importantly, the $\phi$ fluctuations can only be activated at exceedingly low energies/temperatures $\omega, T J/N$ and above that scale their effect can be neglected.

Whenever present, such fluctuations provide "gravitational dressing" of any products of the vertex operators $e^{\phi_i(\tau)}$. This effect can be elegantly carried out with the use of the eigenstates of the exactly solvable Liouville quantum mechanics deformed by the "quench" potential acting between consecutive applications of such operators [49,50].

As the result, an arbitrary power $p$ of the fermion propagator of an isolated SYK system develops a universal asymptotic behavior for all the integer $p$ and $q > 2$

$$< G_i^p(\tau) > \sim N^{3/2 - 2\Delta p}/(J\tau)^{3/2} \tag{12}$$

where the averaging stands for a functional integration over the soft "Schwarzian" modes $f_i(\tau)$ [49,50].

Moreover, if the local reparametrizations were locked into one global transformation $f(\tau)$, thereby drastically reducing the space of the low-energy deformations of the solution (9), then the universal asymptotic (12) would even be shared by the multi-local products $< \prod_i G_i^{p_i}(\tau) >$.

## 4. Phase Fluctuations

The phenomenological action (11) conceived in [9,10] under the customary assumption of a regular gradient expansion does not account for any singular (temporally non-local) effects of the SYK correlations, nor does it allow for a systematic derivation of any non-Gaussian terms.

The classic studies of such effects in the conventional (FL) granular materials were facilitated by representing the fermion operator as a product of its energy-related and charge-related constituents: $\psi_{i\alpha} = \chi_{i\alpha} e^{i\Phi_i}$ [41–44].

Correspondingly, the fermion propagator factorizes

$$G_{ij}(\tau) = \mathcal{G}_{ij}(\tau) D_{ij}(\tau) =$$
$$< \chi_i(\tau)\chi_j^\dagger(0) >< e^{i\Phi_i(\tau)} e^{-i\Phi_i(0)} > \tag{13}$$

onto its "energy" and "charge" components.

The "fractionalized" fermionic degrees of freedom $\chi_{i\alpha}$ can still be traded for the SYK field $\phi$ corresponding to the quasiparticle-hole excitations, while the phase variable $\Phi$ describes the collective ("plasmon") mode.

As already mentioned, at a sizeable charging energy the phase fluctuations dominate in the entire range $J/N < T < E_C$ where the SYK fluctuations remain frozen.

Besides affecting the $\mathcal{G}$ propagator, as per Equations (7) and (8), the tunneling term (3) introduces a (singular) non-Gaussian term into the effective action for the phase field

$$S_{NG}(\Phi) = \frac{1}{2} \sum_{ij} \int_{\tau_1,\tau_2} K_{ij}(\tau_1 - \tau_2) \cos(\Phi_{ij}(\tau_1) - \Phi_{ij}(\tau_2)) \tag{14}$$

where $\Phi_{ij}(\tau) = \Phi_i(\tau) - \Phi_j(\tau)$ and the trigonometric functional dependence stem from the intrinsic compactness of the phase variable subject to the periodic boundary condition, $\Phi_i(\tau + 1/T) = \Phi_i(\tau) + 2\pi n_i$.

The kernel $K_{ij}(\tau) = t^2 \mathcal{G}_i(\tau)\mathcal{G}_j(-\tau)$ in the "influence functional" (14) represents the effect of a dissipative particle-hole bath on the phase dynamics.

On the metallic side of the putative metal–insulator transition and for $T = 0$, this kernel decays algebraically, albeit with different exponents depending on whether or not the system is near criticality.

Deep in the FL phase and away from the critical regime, the phase propagator $D_{ij}(\tau)$ remains nearly constant and the kernel reads

$$K_{ij}(\tau) = \delta_{j,i+\hat{\mu}}(g E_C^{2\epsilon} / \tau^{2-2\epsilon}) \tag{15}$$

where the strength of tunneling is quantified in terms of the dimensionless "conductance." $g \sim t^2 / J^{2-2\epsilon} E_C^{2\epsilon}$.

The time dependence is controlled by the exponent $\epsilon = 1 - 2\Delta$ which varies between 0 (FL, $q = 2$) and 1 ( free dispersionless fermions, $q \to \infty$), thereby making the kernal (15) generically sub-Ohmic for all $q > 2$.

This should be contrasted against the case of an ordinary (FL) granular system where such a regime could only be attained in the presence of a sufficiently strong excitonic enhancement. Otherwise, the kernel (15) would instead turn super-Ohmic due to the competing effect of orthogonality catastrophe [41–44].

At a would-be quantum critical point the system is expected to undergo a transition from the disordered ($< \cos\Phi_i >= 0$, conceivably for $g < g_c$) insulating state governed by the Coulomb

blockade (CB) to a dissipation-driven ordered ($< \cos \Phi_i > \neq 0$) conducting one for $g > g_c$. In the latter state, a condensation of the phase field $D_{ij}(\tau \to \infty) = const$ implies a vanishing effective charging energy $E_C^*$.

## 5. Mean-Field Analysis

In the critical regime, the system of coupled equations for the $\mathcal{G}$ and $D$ propagators reads

$$J^2 \int_\tau [\mathcal{G}]_{ik}^{q-1}(\tau_1 - \tau)\mathcal{G}_{kj}(\tau - \tau_2) +$$

$$t^2 \int_\tau [\mathcal{G}D^2]_{ik}(\tau_1 - \tau)\mathcal{G}_{kj}(\tau - \tau_2) = \delta_{ij}\delta(\tau_1 - \tau_2)$$

$$t^2 \int_\tau [\mathcal{G}^2 D]_{ik}(\tau_1 - \tau)D_{kj}(\tau - \tau_2) = \delta_{ij}\delta(\tau_1 - \tau_2) \tag{16}$$

Incidentally, similar equations and their solutions have been explored in a number of recent works dealing with the mathematically related problem of the transitions between metallic spin-glass and disordered FL states in the randomized Hubbard and $t - J$ models [51–54].

At the critical point, the spatially local and temporally algebraic behavior inherited from the pure SYK model extends all the way down to the lowest energies/temperatures. In particular, the fermion propagator $\mathcal{G}$ retains its SYK behavior (4) with the fermion dimension $\Delta$ while the algebraically decaying phase correlator

$$D_{ii}(\tau) = B/(E_C\tau)^{2\Delta_\Phi} \tag{17}$$

manifests the exponent $\Delta_\Phi = \epsilon/2$.

The dimensionless amplitudes $A$ and $B$ then satisfy the equations

$$A^q + \alpha g B^2 A^2 = 1 \qquad \beta g A^2 B^2 = 1 \tag{18}$$

which allow for a non-trivial solution provided that the numerical prefactors obey the condition $\alpha < \beta$.

Notably, the overall exponent governing the decay of the physical fermion propagator $G$ attains the FL value, $2\Delta_\Phi + 2\Delta = 1$, thereby connecting smoothly with that in the FL phase for $g > g_c$.

Thus, invoking the phase fluctuations appears to be instrumental for reconciling the seemingly conflicting predictions for the fermion dimension $[\psi]$ that one would obtain by approaching the quantum critical point from the FL phase (where $[\psi]_{FL} = 1/2$) by lowering $g$ towards $g_c$ at $T = 0$, as compared to lowering $T$ within the SYK phase (where $[\psi]_{SYK} = \Delta$) at $g = g_c$.

The properties of the critical point can be further discerned by employing a mean-field analysis akin to those of [41–44]. To that end, a two-component $O(2)$ bosonic variable $w_{1,2} = (\cos\Phi, \sin\Phi)$ (or equivalently, one unimodular complex-valued variable $w = w_1 + iw_2 = e^{i\Phi}$) is promoted to a multi-component vector $w_{1,...,M}$ transforming under $O(M)$ and described by the "dissipative non-linear $\sigma$-model":

$$S_{NL\sigma}(\mathbf{w}, \lambda) = \sum_i \int_\tau \left(\frac{1}{2E_C}(\partial_\tau \mathbf{w})^2 + i\lambda(\mathbf{w}^2 - 1)\right)$$

$$+\frac{1}{2}\sum_{ij} \int_{\tau_1,\tau_2} K'_{ij}(\tau_1 - \tau_2)\mathbf{w}_i(\tau_1)\mathbf{w}_j(\tau_2) \tag{19}$$

where the self-consistently determined near-critical kernel differs from (15) due to the extra $\Delta_\Phi$

$$K'_{ij}(\tau) = K_{ij}(\tau)D_{ij}(-\tau) = \delta_{j,i+\hat{\mu}}(gE_C^\epsilon/\tau^{2-\epsilon}) \tag{20}$$

In the $M \to \infty$ limit, the Lagrange multiplier enforcing the local normalization condition $\mathbf{w}_i^2(\tau) = 1$ tends toward a spatially-and temporally-independent value $\lambda_i(\tau) = \lambda$ which can be found from the mean-field integral equation

$$\sum_{\mathbf{q}} \int_{\omega} \frac{1}{\omega^2 / E_C(\mathbf{q}) + zg E_C^\epsilon |\omega|^{1-\epsilon} + i\lambda} = 1 \tag{21}$$

where the propagator of the $\mathbf{w}$-field is read off from (19) and $E_C(\mathbf{q})$ is given by the Fourier transform of the intra/inter-site capacitance matrix $C_{ij}$.

In the FL case ($\epsilon = 0$) the (real-valued) mean-field average $< i\lambda >$ remains finite for all values of the dimensionless parameter $g \sim (t/J)^2$, thus signaling the inescapable onset of the classical CB with a reduced, yet finite, Coulomb gap: $E_C^* =< i\lambda >= E_C \exp(-O(zg))$ and $E_C(1 - O(zg))$ for $zg >> 1$ and $zg << 1$, respectively [41–48].

Qualitatively, this insulating behavior persists for all $\epsilon < 0$ where the kernel (20) is super-Ohmic as, e.g., in the universal regime (12), which if applicable, would formally correspond to $\epsilon = -1/2$.

In contrast, for $0 < \epsilon < 1$ the integral (21) remains finite even in the limit of $\lambda \to 0$, thanks to the sub-Ohmic dissipative term. It then gives rise to a finite critical conductance

$$g_c = 1/(z\epsilon^{1+\epsilon}) \tag{22}$$

above which $\lambda = 0$, thereby signaling a quenching of CB and onset of a metallic behavior. In the ordinary FL granular materials, such a behavior could only occur in the presence of sub-Ohmic dissipation due to either a coupling to external bath or intrinsic excitonic effects [41–44].

In terms of the critical tunneling amplitude, the transition occurs at $t_c \approx J^{1-\epsilon} E_C^\epsilon / (z\epsilon^{1+\epsilon})^{1/2}$ and its only dependence on the lattice structure is through $z$. Upon approaching the FL ($\epsilon \to 0$) the transition becomes unattainable.

Additionally, in the customary case of $q = 4$, said transition takes place at $t_c \sim (JE_C)^{1/2}$ (or equivalently, $E_C \sim t^*$), in agreement with the earlier conclusions drawn for the SYK-lattices [21–35].

Upon moving deeper into the insulating phase the renormalized Coulomb (Mott) gap rises, as dictated by Equation (21)

$$E_C^* = E_C(1 - g/g_c)^\nu / \epsilon^2 \tag{23}$$

with the critical exponent $\nu = (1 - \epsilon)/\epsilon$.

Notably, for $q = 4$ the gap scales linearly with a deviation from the critical point while for $q \to \infty$ the gap emerges abruptly and the transition resembles that of first order.

## 6. Conductivity

The charge transport properties of a granular array can be assessed by computing the conductivity

$$\sigma_{\mu\nu}(\omega) = \frac{ia^{2-d}}{\omega} \int_{\tau} e^{i\omega\tau} (\Pi_{\mu\nu}^{dia}(\tau) + \Pi_{\mu\nu}^{para}(\tau))|_{\omega \to -i\omega + 0^+} \tag{24}$$

where $d$ is the spatial dimension.

The diamagnetic and paramagnetic contributions towards the overall conductivity read

$$\Pi_{\mu\nu}^{dia}(\tau) = \delta_{\mu\nu} \frac{g}{\pi} \int_{\tau'} (\delta(\tau) - \tau\delta(\tau - \tau'))$$
$$\sum_{\rho} K(\tau') < \cos(\Phi_{i,i+\rho}(\tau) - \Phi_{i,i+\rho}(\tau')) > \tag{25}$$

and

$$\Pi_{\mu\nu}^{para}(\tau) = \frac{g}{\pi} \int_{\tau',\tau''} K(\tau - \tau') K(\tau'') \tag{26}$$

$$< \sin(\Phi_{i,i+\mu}(\tau) - \Phi_{i,i+\mu}(\tau')) \sin(\Phi_{i,i+\nu}(0) - \Phi_{i,i+\nu}(\tau'')) >$$

As in the Ohmic case [45–48], one can show that the dominant contribution comes from the 1st order diamagnetic term while the corresponding second order correction cancels against the paramagnetic one.

Besides, in contrast to the case of a single junction where the dominant (albeit subleading, $\sim g^2$) contribution towards the low-$T$ conductance is provided by inelastic co-tunneling processes [11–15], the latter appear to be suppressed exponentially with the size of the array [45–48].

Keeping the diamagnetic term, one then arrives at the formula

$$\sigma_{\mu\nu}(T) = a^{2-d} \sum_{\mathbf{q}} \int_\omega \frac{1}{\omega} \frac{\partial n(\omega)}{\partial \omega} \mathbf{s}_q^\mu \mathbf{s}_q^\nu$$

$$\int_\tau K(\tau)(1 - \cos\omega\tau) e^{-W(\tau)} \tag{27}$$

where $\mathbf{s_q} = \partial_q c_{\mathbf{q}}$ is a gradient of the sum over the nearest neighbors $c_{\mathbf{q}} = \sum_{\hat\mu}(1 - e^{i\mathbf{q}\hat\mu})$, and the Debye–Waller (DW), which stems from the Gaussian averaging of the exponentials of the phase field given by the exponential of

$$W(\tau) = \sum_{\mathbf{q}} \int_\omega \mathbf{s_q}^2 (1 - \cos\omega\tau) < |\delta\Phi(\omega, \mathbf{q})|^2 > \tag{28}$$

Computing (27) one finds an approximate, yet practically convenient expression for the longitudinal conductivity in terms of the Fourier transform $\tilde K(\omega)$ of the kernel (15)

$$\sigma(T) \sim e^{-W(1/2T)} \tilde K(T)/T \tag{29}$$

proposed "ad hoc" in the early work of [41–44].

Away from criticality, the phase fluctuations' propagator entering the DW factor (28) reads

$$< |\delta\Phi(\omega, \mathbf{q})|^2 >= \frac{1}{\omega^2/E_C(\mathbf{q}) + gE_C^{2\epsilon}|\omega|^{1-2\epsilon}c_{\mathbf{q}}} \tag{30}$$

In the FL case ($\epsilon = 0$), the diffusion term $\mathcal{D}\mathbf{q}^2$ appearing in Equation (11) derived by virtue of a phenomenological gradient expansion can be identified with (and absorbed into) that proportional to the conductance $g$, whereas for $\epsilon > 0$ it can be neglected altogether compared to the (singular) latter term.

It is worth pointing out that the momentum sum in (28) turns out to be non-singular even in the potentially problematic dimensions $d = 1$ or 2, the only information about the lattice being its coordination number.

In the deep CB regime corresponding to $g \ll g_c$ one might need to keep track of the large phase field fluctuations when computing the DW factor

$$< e^{i\Phi_i(\tau)} e^{-i\Phi_i(0)} >= e^{-\frac{1}{2}E_C^*\tau} \sum_n e^{-\frac{E_C^*}{2T}n(n+2T\tau)+2\pi n\mathcal{E}} \tag{31}$$

where the infinite sum over the winding numbers restores the periodicity under $\tau \to \tau + 1/T$.

The potential importance of the large phase fluctuations (hence, non-trivial winding numbers) brings about a conductance dependence on the offset charges $Q_i$. Their effects can be studied by

restoring the topological "$\theta$-term" originating from the cross-terms in the charging energy (4), $S_{top} = i\sum_i (Q_i - 2\pi T\mathcal{E}/E_C)\int_\tau \partial_\tau \Phi_i$, which accounts for the $T$-dependence of the intrinsic excess charge on the dots through the relation $2\pi\mathcal{E} = -\partial\mu/\partial T|_{T\to 0}$ [1–10].

In that regard, our discussion pertains to the charge quantization plateaus $Q_i = n$ where the (renormalized) Coulomb gap is maximal and the system is least likely to go metallic. In contrast, at the transition points between the plateaus ($Q_i = n + 1/2$) where the bare gap $E_C^* = E_C(1 - 2 < Q_i >)$ vanishes, the conductivity takes its maximal values. The discussion of such a (near) degenerate regime will appear elsewhere.

Nonetheless, at low $T$ the non-trivial winding numbers can be neglected, and for $gE_C^* < T < E_C^*$ the conductivity governed by the $n = 0$ term in the sum (31) shows the ordinary Arrhenius behavior

$$\sigma(T) \sim \sigma_0 \exp(-E_C^*/T) \tag{32}$$

where $\sigma_0 \sim a^{d-2}g$. For $\epsilon = 0$ the insulating behavior sustains at all $g$.

As tunneling increases or temperature decreases, $T < gE_C^*$, Equation (29) yields

$$\sigma(T) \sim \sigma_0 (T/gE_C^*)^{1/\pi zg} \tag{33}$$

Expanding the DW factor to 1st order reproduces the (negative) logarithmic (in all dimensions) conductivity correction, $\sigma(T)/\sigma_0 = 1 - O(1/zg)\ln(gE_C^*/T)$.

Interestingly enough, the above result appears to be accurate to the next, second order due to the aforementioned cancellation between the higher order diamagnetic and paramagnetic corrections [45–48].

As temperature decreases, the (negative) logarithmic conductivity correction gets cut off at energies $\sim g\delta$ (the rate of fermion escape from a dot) and becomes comparable to the bare conductivity for $g \sim (1/z)\ln(E_C^*/\delta)$, again in agreement with the results of [45–48].

In contrast, for $\epsilon > 0$ the conductivity suppression due to the DW factor remains non-singular at $T \to 0$ and Equation (29) demonstrates the NFL power-law

$$\sigma(T) \sim \sigma_0 e^{-W(0)}(E_C^*/T)^{2\epsilon} \tag{34}$$

governed by a generically non-integer exponent.

Incidentally, though, for $q = 4$ Equation (34) features a linear resistivity, consistent with the experimental data on a variety of the prospective SM compounds [36–40].

However, with increasing temperature the DW factor starts to contribute as well, resulting in a competition between the "kinematic"'s power-law (34) dictated by the SYK propagator (9) and the fractional-exponential $T$-dependence of the correlation-induced $W(1/2T)$

$$\ln\sigma(T)/\sigma_0 = (T/E_C^*)^{2\epsilon}/(2\epsilon zg) - 2\epsilon\ln T/E_C^* \tag{35}$$

The conductivity behavior switches from decreasing to growing, as signified by the sign change of $\ln\sigma(T)$, at $T = t^* = (2\epsilon z)^{1/2\epsilon}t^{1/\epsilon}J^{1-1/\epsilon}$, consistent with the previously quoted value of $t^*$ for $q = 4$ [21–35].

## 7. Density of States

Another important marker of the metal-insulator transition is a concomitant "zero-bias anomaly" in the fermion density of states (DOS). By evaluating the latter with the use of the factorization formula (13), one obtains

$$\nu(\omega) = \frac{1}{\pi}Im\int_\tau e^{i\omega\tau}\mathcal{G}(\tau)e^{-W'(\tau)}|_{\omega\to -i\omega+0} \tag{36}$$

This time around the DW factor stands for the average of only two (rather then four, as in Equation (27)) exponentials of the phase field

$$W'(\tau) = \sum_{\mathbf{q}} \int_{\omega} (1 - \cos \omega \tau) < |\delta\Phi(\omega, \mathbf{q}|^2) > \tag{37}$$

Additionally, as opposed to the momentum sum in Equation (28), its counterpart (37) appears to be rather sensitive to the spatial dimension, whose dependence is not limited to that on (and, in fact, does not involve) the coordination number $z$.

In particular, for $d = 2$ the momentum sum is logarithmic, thereby reproducing the log-normal "zero-bias anomaly" familiar from the general theory of 2$d$ disordered conductors [45–48] in the FL case ($\epsilon = 0$)

$$\nu(\omega) \sim \frac{1}{J} \exp(-\frac{1}{\pi g} \ln^2(g E_C^*/\omega)) \tag{38}$$

The lack of information about the lattice in Equation (38) can be understood from the fact that the momentum sum in (37) is dominated by small (rather than large, as in (28)) momenta.

By comparison, for a generic $\epsilon > 0$ and $d = 2$ one obtains the tunneling DOS.

$$\nu(\omega) \sim J^{-1}(\omega/J)^{(1/\epsilon g^{1/(1+\epsilon)})-\epsilon}(J/t)^{O(1/\epsilon^2 g)} \tag{39}$$

For $g >> 1$ and $q = 4$ Equation (39) behaves as $1/\omega^{1/2}$, reproducing the salient $SYK_4$ transport feature [11–15].

The overall sign of this power-law dependence changes from negative (SM) to positive (MI) at the critical conductance $g_c' = 1/\epsilon^{2(1+\epsilon)}$ which appears to be generally consistent with (22).

Instead, for $d \geq 3$ the momentum sum in (37) becomes non-singular, thereby making the DW factor finite at all $\tau$ and resulting in a generic linear DOS for $\omega \ll E_C^*$:

$$\nu(\omega) \sim \omega/J^{1-\epsilon}(E_C^*)^{1+\epsilon} \tag{40}$$

thereby showing the development of a "soft" gap.

The latter is markedly different from both the hard gap $\nu(\omega) \sim \theta(\omega - E_C^*)$, which is a hallmark of the CB in a FL with momentum-dependent dispersion, and the bare DOS of the degenerate species, $\nu(\omega) \sim \delta(\omega - E_C^*)$.

Thus, by measuring the tunneling DOS one might be able to access the properties of the physical fermion propagator across all the different regimes. Overall, its evolution with energy/temperature can be summarized as follows.

At $\omega, T \gg J$ it is that of free dispersionless fermions, $G(\tau) \sim sgn(\tau)$, which corresponds to the bare fermion dimension $[\psi]_0 = 0$ under the time dilation ($\tau \to l\tau$). However, as the scale drops below $J$ and the system enters the SM regime it evolves towards the SYK mean-field value $[\psi]_{SYK} = \Delta$.

Further, once the systems cools down to $Tt^*$, the strongly relevant tunneling term continues to monotonically drive the dimension from the SYK value $\Delta$ towards the FL one, $[\psi]_{FL} = 1/2$. For $g > g_c$ (or equivalently, $E_C^* < t^*$) the SM gives way to a disordered FL, whereas for $g < g_c$ (or $E_C^* > t^*$) one expects a transition to the MI.

## 8. Discussion

The above scenario of the MIT in a granular SYK array can be viewed as being somewhat complementary to that presented in the recent [55]. Rather than the CB effects, that work was mainly concerned with the effects of the Schwarzian fluctuations.

On the technical side, the renormalization group (RG) equations derived in [55] contain a conveniently chosen scale-dependent fermion dimension $[\psi](l)$. In the standard RG procedure, though, $[\psi]$ should have instead been found from the corresponding fermion field renormalization

factor—which, in turn, would have to be computed as a sole function of the independently determined dimensionless RG charges, obeying their own closed system of equations.

Besides, the MIT studied in [55] occurs at the low tunneling strength, $t \sim J/N$, thereby implying that for $N \gg 1$ and finite temperatures the system behaves as a metal for all the practical purposes.

Compared to the above result, our analysis focuses on the role of the charging effects and predicts the onset of metallic behavior in the SYK array upon increasing the tunneling strength (or equivalently, the inter-site conductance $g$) past the $N$-independent threshold value (see Equation (22)).

While being, at first sight, similar to the observations made in [21–35], our findings appear to be starkly different, as far as both the underlying mechanism and the actual critical parameter values are concerned. Besides, the proposed scenario has no analogue in the case of the FL granular system without an additional source of sub-Ohmic dissipation.

It should be noted, though, that the standard influence functional approach used in this (as well as much of the previous) work is only applicable when all the relevant energy/temperature scales—$J, t^*, E_C^*$, etc.—exceed the average single-particle level spacing $\delta \sim J/N$ (moreover, its many body counterpart, $\delta_N \sim J \exp(-O(N))$ [1–3,49,50]).

Incidentally, at energies of order $\delta$ the renormalizing effects of the Schwarzian fluctuations would have just started to develop and the universal regime (12) (let alone the MIT scenario of [55] could not have yet been reached.

Furthermore, as recently shown in the case of a single tunnel junction [15], at such low energies one might expect an intricate competition between the SYK, charging, tunneling, and single and multi-level Kondo phenomena. Therefore, for a complete picture it might be necessary to consider the charging and SYK effects on equal footing with the potentially important local Kondo resonances.

Lastly, by assuming the simplest nearest-neighbor tunneling, we deliberately left out such subtle topics as variable range hopping (Mott, Efros–Shklovskii, and related mechanisms, all capable yielding $\sigma(T) \sim \exp(-(E_C^*/T)^\nu)$ with various fractional exponents $\nu$) whose inclusion is likely to be necessary if a detailed comparison with experimental data on any actual SYK arrays were ever to be made.

Considering the long and still unfinished history of the studies of the phenomenon of CB even in the ordinary FL granular materials, it would be rather unrealistic to try to cover a potentially rich variety of the pertinent regimes all at once. One might hope, however, that the present attempt to shed some light on the new aspects of this long-standing problem will revitalize the field as a whole.

**Funding:** This research received no external funding.

**Acknowledgments:** The author acknowledges the hospitality at and support from the Pauli Center for Theoretical Studies (ETH, Zurich).

**Conflicts of Interest:** The authors declare no conflict of interest.

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
