# Peer review of "Connecting the SYK Dots"

_condensedmatter, doi:10.3390/condmat5020037_

Round 1

Reviewer 1 Report

The author presents an interesting theoretical study concerning SYK array of quantum dots and exploring the dependence of the conductivity and density of states in function of the inter-site tunneling strengh.

The manuscript is well written, the results are sound obtaining analytical derivations of expressions previously taken a priori in related works. 

My only criticism regards the excessive use of multiple references  to each citation item which implies to some misleading. I suggest to the author to put a single reference in each citation item and eliminate those works which are not relevant to the discussion. 

There are also some typos:

Line 103: the comma is misplaced in the beggining of the statement;

Line 127: "conductance" is mispelled ("çonductance")

Line 156: "extra" is mispelled.

Line 258: remove the comma before "etc"

Reviewer 2 Report

The author theoretically studied the SKY model.

However, the following issues are unclear.

  1. The importance and the conclusions of this study.
  2. Which and how parameters cause the MIT in dots. Illustrations can help the changes of DOS and charge carrier density in conduction band
  3. Practical applications and comparison of other MIT models, such as Mott-Hubbard transition.

Author Response

The comments of Reviewer 1 (which are similar to some of those made by Reviewer 2) are taken into account in the updated version of the manuscript where the importance of this study, its motivation, originality, and the conclusions  are repeatedly spelled out in the Abstract, Introduction, and Discussion. 

In particular, it is emphasized that the parameter driving the metal-insulator transition is the inter-dot conductance, akin to that in the theory of Coulomb blockade in the ordinary (Fermi liquid-like) granular materials.   

It is also mentioned that while no practical realizations of the SYK arrays are currently available, the lack of such systems has neither been viewed as an argument against studying this novel system, nor has it prevented a flurry of the recent theoretical studies on this and related topics from being published  (for a by far incomplete list see Refs.[21-41, 56]). 

As regards the other known types of the metal-insulator transition, the most closely related counterpart of the mechanism discussed in the manuscript is that predicted to operate in the ordinary (Fermi liquid-like) materials in the presence of a sub-Ohmic  dissipative bath. While belonging to the general family of the repulsive interaction-driven Mott-Hubbard phenomena the important specifics of this transition (namely, the concrete dependencies presented in Eqs.(22,23,33,34,35,39,40)) put them in a league of their own. We believe that presenting such results makes the comparison requested by the Reviewer self-evident.     

Reviewer 3 Report

The manuscript “Connecting the SYK Dots” discusses the particular metal-to-insulator transition in a granular array of the 2 SYK (Sachdev-Ye-Kitaev) quantum dots.

The manuscript is well written and the scientific level appropriate and reflect the current state of the art, and proposed several new approaches considering the interaction of Sachdev-Ye-Kitaev quantum dots.

The Abstract should contain also the main conclusions of the study. Also, the main conclusions are generally lacking at the end of the manuscript.

Several hypothetical applications and practical consequences of such particular interaction of SYK-QD should be provided (at least at the hypothetical level).

Also, the material should be better organized by the introduction of subtitles when appropriate. Although the manuscript is logically drafted, this will make the current version more comprehensible and reader-friendly.

The originality and the contribution to the advancement of the field should be clearly stated.

The references should be outlined appropriately. The current form cannot be accepted.

Author Response

We appreciate the Reviewer 2's assessment of the manuscript as being well written and its scientific level as being appropriate, novel, and reflecting the current state of the art.

In response to the Reviewer's critical remarks we made significant changes to the text of the manuscript, spelling out our motivations and main conclusions (in the Abstract, Introduction, and Discussion), points of departure from the earlier works, as well as the potential manifestations of the proposed scenario of the metal-insulator transition in any future realizations of the SYK arrays.

It should be noted, though, that a large number of the previously published papers on the related topics (Refs.21-36) would not even consider any practical implementations of the hypothetical SYK-lattices, while other works (Refs.37-41) would only speculate about the possibility of applying their formal results to such 'usual suspects' among the concrete experimental systems  as the cuprates and heavy-fermion materials.  Although some proposals for constructing the single SYK quantum dots were indeed made (Refs.16-20), their practical realizations (let alone building an extended array of such SYK dots) have not yet been attempted.  

Also, we heeded to the Reviewer's advice to subdivide the manuscript to the sections under the individual subtitles. We believe that it made  the updated version more comprehensible and reader-friendly. Lastly, as per the Reviewer's request, we separated out the previously bundled multiple references.

Reviewer 4 Report

The author investigates properties of a granular SYK model taking into account Coulomb blockade effects. The Coulomb interaction has been considered for the SYK model for the first time. Proper techniques borrowed from theory of granular metals an insulators has been used and the results are reliable.

The paper can be of interest to researchers working on various SYK-models and other field theories. I recommend the paper for publication.       

Author Response

The thank Reviewer 3 for endorsing the manuscript, as far as its novelty,  methods, and reliability of the results are concerned. There are no critical points to address.   

Round 2

Reviewer 2 Report

The author theoretically examined the MIT of the SYK dots for N>>1.
It is quite interesting result. For N>>1 cases, the boundary effect may be less important.
However, the effects of boundaries and defects in quantum dots cannot be ignored in reality.
I am not sure if this calculations can include the effects of the boundaries and defects.